# Pediatric Episodic Migraine with Aura: A Unique Entity?

**DOI:** 10.3390/children8030228

**Published:** 2021-03-17

**Authors:** Hannah F.J. Shapiro, Alyssa Lebel

**Affiliations:** 1Department of Child Neurology, Boston Children’s Hospital, Boston, MA 02115, USA; Hannah.johnson@childrens.harvard.edu; 2Division of Pain Medicine, Department of Anesthesiology, Boston Children’s Hospital, Boston, MA 02115, USA

**Keywords:** migraine, migraine with aura, brainstem aura, hemiplegic migraine, retinal migraine, pediatric migraine

## Abstract

Migraine headache is a common cause of pain and disability in children and adolescents and is a major contributor to frequently missed school days and limitations in activities. Of children and adolescents with migraine headache, approximately one-third have migraine with aura (MA). MA is often considered to be similar to migraine without aura (MO), and thus, many studies do not stratify patients based on the presence of aura. Because of this, treatment recommendations are often analogous between MA and MO, with a few notable exceptions. The purpose of this review is to highlight the current evidence demonstrating the unique pathophysiology, clinical characteristics, differential diagnosis, co-morbidities, and treatment recommendations and responses for pediatric MA.

## 1. Introduction

Migraine headache is a common cause of pain and disability in children and adolescents and is a major contributor to frequently missed school days and limitations in activities [1,2,3]. The prevalence of migraine, including migraine with aura, in pre-pubertal children is approximately 5% and increases throughout adolescence and young adulthood to a peak of approximately 30% around age 25 [4,5,6]. Migraine is considered a life-long disorder since approximately 60% of children diagnosed with migraine with or without aura continue to have migraine attacks during adulthood [7]. Of children and adolescents with migraine headache, approximately one-third have migraine with aura (MA) [5,6,8]. Aura is a heterogenous and complex entity that shares clinical features with more sinister neurological conditions. While MA and migraine without aura (MO) are often considered to be similar headache disorders, MA may have different pathophysiology compared to MO, leading to several unique features. The purpose of this review is to highlight the current evidence demonstrating the unique pathophysiology, clinical characteristics, differential diagnosis, co-morbidities, and treatment recommendations and responses for pediatric MA. Although there is significant overlap between the diagnosis and treatment of MA and other primary and secondary headache disorders, this review does not provide a comprehensive review of these alternative diagnoses.

## 2. Pathophysiology

MA is understood to be due to cortical spreading depolarization (CSD) in which depolarization of cortical neurons and glia slowly spreads across the cortex. This depolarization is followed by hyperpolarization, causing large current shifts and release of vasoactive substances. In response to the increased energy demand needed to restore homeostasis, there is an increase in regional cerebral blood flow [9,10]. CSD as the mechanism underlying aura is supported by functional magnetic resonance imaging data demonstrating blood oxygenation level-dependent signal changes progressing over the visual cortex during visual aura [11]. There are several proposed mechanisms to explain how CSD leads to the headache phase of MA, including CSD triggering an inflammatory cascade and releasing nociceptive substances and CSD directly activating the trigeminocervical complex [12,13]. While the exact pathophysiology of MO is less understood, there are data to suggest that the pathophysiology is different from MA. Perhaps the most convincing evidence is that inhibiting CSD using tonabersat prevents attacks of MA but has no effect on attacks in MO [14]. Additionally, imaging studies have suggested differences in cerebral blood flow and cortical thickness between patients with MA and MO [15,16].

Translational research over decades has proven the critical role of calcitonin gene-related peptide (CGRP) in MA pathophysiology, though it is still uncertain whether there is a differential importance in MA compared to MO. CGRP is a neuropeptide abundant in trigeminal ganglion neurons involved in pain signaling [17]. Trigeminal activation may induce release of CGRP from the peripheral terminals, triggering a cascade of events leading to migraine attacks [18,19]. In animal models, there is evidence that CGRP levels increase during CSD and that blocking CGRP reduces CSD [20]. In adults, CGRP triggers migraine attacks in patients with typical aura and without aura, suggesting CGRP may play a key role in both migraine disorders regardless of the presence of aura [19,21]. There are some data to suggest that CGRP does not trigger migraine attacks in certain subsets within MA, specifically familial hemiplegic migraine, suggesting that perhaps not all subsets of MA have the same pathophysiology [22]. In pediatric migraine patients, plasma levels of CGRP are elevated during migraine attacks, suggesting that CGRP likely plays an important role in pediatric migraines as well [23]. However, additional studies are needed that stratify pediatric patients by the presence of aura to better understand the role of CGRP in pediatric patients who have MA.

## 3. Clinical Characteristics and Diagnostic Criteria

The International Classification of Headache Disorders (ICHD) distinguishes MA from MO. The defining feature of MA is the presence of aura, which may be present without a headache. When head pain is present, aura typically precedes it, but both features can occur simultaneously [21]. Compared to MO, patients with MA tend to have fewer attacks per month, but the episodes tend to be more severe [5]. Children with MO may be diagnosed with MA as they age, perhaps in part because some children (e.g., young children) may struggle to articulate the symptoms of aura. As a result, it is important to take frequent detailed headache histories in pediatric patients [24]. Aura is classified into subtypes and must satisfy specific criteria related to duration, laterality, and symptom quality. The current diagnostic criteria from the ICHD-3 for MA are outlined in Table 1 [25]. These criteria are the same for adult and pediatric patients. Within MA, there are four distinct types: migraine with typical aura, migraine with brainstem aura, hemiplegic migraine, and retinal migraine. It is important to highlight the unique features of hemiplegic migraine and migraine with brainstem aura as these are often thought of as more severe forms of migraine which has implications for treatment recommendations.

### 3.1. Migraine with Typical Aura

Migraines with visual aura, hemiparaesthetic aura, and speech/language aura are classified as migraines with typical aura. Of these, visual aura is the most common type of aura followed by sensory, then speech and/or language [5,6,26]. A typical visual aura is characterized by unformed black and white patterns (e.g., zigzags, crescents, flickering) that first appear in the center of the visual field and then move to the periphery, followed by a scotoma [26]. However, there is substantial variability in the characteristics of visual auras [27]. For example, in children and adolescents, polychromatic aura and formed shapes (e.g., dots, circles, triangles, squares, stars) are often reported [8]. Sensory aura is classically characterized by a unilateral “marching” phenomenon, starting with tingling in the hand and spreading up the arm to the face, sometimes involving the buccal mucosa and tongue. Often the tingling is followed by numbness. However, numbness may be the only symptom [26]. Speech and/or language auras are usually described as aphasias but may also include slowed speech or difficulty reading, especially in adolescents [26,28].

In a migraine with typical aura, visual aura is almost always present. However, in 2–10% of migraine aura attacks, nonvisual aura symptoms occur in isolation [5,29,30]. When multiple types of typical auras are present in a single attack, these auras can present simultaneously or in succession [31]. While the majority of pediatric patients report a gradual development of aura over at least 5 min, a minority of patients do develop aura faster (<5 min) [8]. Additionally, single aura symptoms in this population may last <5 min or longer than 60 min [8,30,31,32,33,34,35]. Given the variability in aura features, it is important to recognize that only three of the six aura characteristics outlined in the ICHD-3 criteria are needed to diagnose MA.

### 3.2. Migraine with Brainstem Aura

Migraine with brainstem aura is diagnosed when patients have MA together with at least two fully reversible brainstem auras: dysarthria, vertigo, tinnitus, hypacusis, diplopia, ataxia, and/or decreased level of consciousness [25]. Motor symptoms and retinal symptoms must not be present. Vertigo is the most commonly reported brainstem aura [36]. Nearly all patients with a brainstem aura also have a visual aura, which usually precedes the brainstem aura [37].

### 3.3. Hemiplegic Migraine

Motor aura is classified as a hemiplegic migraine (HM). While the hallmark of HM is fully reversible motor weakness, the motor symptoms are always accompanied by at least one other type of aura (i.e., visual, sensory, speech/language). Thus, the diagnostic criteria for HM include the criteria for MA in addition to fully reversible motor weakness. HM has a number of features that make this disorder distinct compared to other migraine subtypes. Specifically, a genetic basis has been identified in at least a portion of these cases. HM can be sporadic or familial, and the associated gene mutations are thought to cause channelopathies [38]. While there is variability reported in the literature, the duration of motor symptoms tends to be longer compared to other auras [39]. In the ICHD-3, it is noted that motor symptoms generally last less than 72 h but can persist for weeks [25].

### 3.4. Retinal Migraine

Patients with retinal migraine meet the diagnostic criteria for MA and also have fully reversible, monocular, positive, and/or negative visual phenomena (e.g., scintillations, scotomata, or blindness) confirmed during an attack by either or both of the following: clinical visual field examination and/or the patient’s drawing of a monocular field defect (made after clear instruction) [25]. Additionally, at least two of the following must be present: symptoms spreading gradually over ≥5 min, symptoms lasting 5–60 min, and/or accompanied—or followed within 60 min—by headache. Retinal migraine is exceptionally rare among pediatric migraine patients, accounting for <2% of migraines [36,40]. It is difficult for pediatric patients to differentiate between monocular aura and homonymous hemianopsia aura, making this diagnosis challenging.

## 4. Differential Diagnosis

Because focal neurological signs are present in MA, there are important alternative diagnoses to consider including transient ischemic attack (TIA), stroke, and epileptic seizure. Accurately diagnosing MA may prevent unnecessary testing and may prevent patients from being treated with harmful medications that target other conditions. However, inappropriately diagnosing MA when the correct diagnosis is TIA, stroke, or seizure may lead to delays in initiating life-saving therapies.

### 4.1. Key Differentiating Characteristics

Particular attention to the onset, duration, offset, and quality of symptoms can help to distinguish MA from TIA, stoke, and seizure. A migraine aura has a gradual onset, developing over at least five minutes. TIA, stroke, and seizure all have a sudden onset. Each migraine aura symptom typically lasts for 5–60 min, which can be similar to the duration of TIA symptoms which last from minutes to 24 h. This duration is different compared to stroke symptoms, which are persistent, and seizure symptoms, which are typically <5 min. Offset is gradual in migraine auras but abrupt in TIAs and seizures. A migraine aura typically has positive symptoms (e.g., shimmering, flashing lights, visual hallucinations, paraesthesias) followed by negative symptoms (e.g., scotoma, numbness, weakness), whereas TIA and stroke usually have only negative symptoms. Seizures typically have positive symptoms—similar to migraine auras. For example, occipital seizures can produce positive visual symptoms, often described as colored shapes, similar to visual auras. Because of this, the important distinguishing features in seizures are the much shorter duration of symptoms (<5 min) and the abrupt onset and offset [41].

### 4.2. Specificity of the ICHD Criteria

The detailed nature of the ICHD criteria helps to distinguish MA from other entities based on clinical history. The current version of the ICHD-3 criteria was updated from the ICHD-3 beta criteria in 2018 to better distinguish MA from TIA [42]. The update included the addition of “at least one symptom is positive” to the criteria, as positive symptoms are much more common in migraine aura and negative symptoms are more common in TIA and stroke. There is recent evidence in adults that this updated criterion is significantly better at distinguishing MA from TIA [43]. Since TIA is rare in children, additional studies in children are needed to establish the specificity of the new diagnostic criteria in this population. Prior studies examining the specificity of the ICHD-3 beta criteria in pediatric patients are reassuring as the criteria provide adequate flexibility to capture the heterogenous nature of aura, thus enabling accurate diagnosis in most pediatric patients with aura [8,44]. In future studies, the specificity of the updated criteria should be examined against more common pediatric migraine mimickers, including epileptic seizures, since seizures often produce positive symptoms.

## 5. Co-Morbidities

In adults, MA is associated with a two-fold increase in ischemic stroke [45]. Although the data in pediatric patients are less robust, a large retrospective study found that there is an elevated risk of ischemic stroke in adolescents with MA but not in prepubertal children [46]. This highlights that there is likely a hormonal effect underlying the association between MA and stroke. This idea is corroborated in adult studies in which MA was an independent risk factor for stroke in women of reproductive age [47]. Furthermore, in young women who have MA and use combined hormonal contraceptives, there is a six-fold increase in the risk of ischemic stroke compared to those with neither risk factor [48]. This has significant implications for adolescent girls with MA who are considering options for contraception.

## 6. Treatment Recommendations

Despite the known differences in pathophysiology, clinical characteristics, and co-morbidities in MA compared to MO, the treatment recommendations are almost always the same between these two groups. This is largely due to the fact that research studies typically do not stratify patients by the presence of aura [49]. Because of this, the strongest evidence for treatment of pediatric MA is derived from studies that group MA with MO. Treatment recommendations for both MA and MO include a combination of lifestyle modifications, nonpharmacological interventions, acute therapies, and preventative therapies [50,51].

Counselling on lifestyle modifications is always the first intervention [50,52]. While childhood and adolescence are marked by frequent change, it is vital to educate families on the importance of lifestyle regularity for headache management. In the landmark study Childhood and Adolescent Migraine Prevention (CHAMP), a randomized controlled trial of preventative medications versus placebo for pediatric migraine, approximately 60% of patients in the placebo arm of the study had reduction in headache frequency [53]. It is hypothesized that this exceptional response in the placebo group was due in part to the fact that lifestyle counseling was performed at monthly clinic visits [54]. Counselling should include the importance of maintaining a regular sleep schedule, ensuring adequate sleep duration, eating regular meals, engaging in exercise, increasing fluid intake, and learning stress-management skills [55,56,57,58,59]. Nonpharmacological interventions include cognitive behavioral therapy (CBT), biofeedback, acupuncture, massage, and physical therapy [60,61,62,63,64].

Guidelines from the American Academy of Neurology (AAN) and the American Headache Society (AHS) for the acute treatment of migraines in pediatric patients recommend use of nonsteroidal anti-inflammatory drugs, acetaminophen, triptans, and anti-emetics [50]. It is important to counsel on not using analgesics more than 14 days per month or triptans more than 9 days per month in order to avoid medication-overuse headache [50,65,66]. Because there is insufficient evidence to support the use of specific preventative medications in children—partially due to the high placebo rate seen in this population—if the decision is made to start preventative therapy, often the choice of medication is guided by patient age, the presence of co-morbidities, and clinical experience [51,67]. Commonly prescribed preventative medications include topiramate, amitriptyline, propranolol, and cyproheptadine [51,53,68,69,70,71,72].

## 7. Differences in Treatment Responses

When MA is examined specifically, there is some evidence that it may respond differently to certain treatments compared to MO [73].

### 7.1. Treatment Differences: Acute Therapies

Triptans are serotonin 1b/1d agonists specifically designed to treat acute migraine attacks. There are currently four triptans approved by the United States Food and Drug Administration (FDA) for use in the pediatric population. Rizatriptan is approved for use in children six years of age and older, and almotriptan, zolmitriptan, and sumatriptan/naproxen are approved for use in children aged 12 and older. In adult studies, sumatriptan was found to be less effective for patients who have MA compared to patients with MO [74]. There is insufficient data in pediatric patients examining this finding. However, in a small retrospective review of pediatric patients, the presence of aura was associated with lower likelihood of benefit from triptans [75]. However, it is still recommended to use triptans in pediatric patients with aura, as the majority of these patients benefit from this class of medications.

Magnesium is often used for abortive headache therapy, more often in adults and older adolescents compared to children. In an adult study, intravenous magnesium sulphate significantly reduced head pain and aura in patients with MA, but there was no significant reduction in head pain in patients with MO [76]. There is some evidence in animal models that lower cortical magnesium levels can induce CSD, perhaps suggesting an underlying mechanism [77]. There are yet to be studies examining this finding in pediatric patients.

Transcranial Magnetic Stimulation (TMS) is a non-pharmacological, non-invasive treatment based on the idea that a pulse of magnetic stimulation can disrupt CDS. In adults, there are data to support its use as an abortive option, and it shows better efficacy in adults who have MA compared to MO [78]. There are little data regarding TMS in pediatric patients with migraine. However, in a pilot study, there was evidence that TMS was well-tolerated in adolescent patients with chronic migraines and decreased headache frequency, though only one participant reported MA [79].

### 7.2. Treatment Differences: Preventative Therapies

In adults with MA, there are some studies demonstrating that lamotrigine reduces the frequency of migraine aura and migraine attacks [80,81,82]. However, there are conflicting studies where lamotrigine was not effective at reducing MA [83,84]. In pediatric patients, evidence is insufficient, but there was one small study and one case report showing lamotrigine reduced aura and MA frequency [85,86]. In an animal model, lamotrigine has demonstrated suppression of CSD, which could perhaps explain the mechanism by which this medication reduces MA [87]. However, there are some data arguing against this theory. For example, chronic use of other typical preventative medications that suppress CSD, including propranolol, amitriptyline, and topiramate, do not have convincing data that they are more efficacious in MA, though this hasn’t been systemically studied [88]. In one study examining topiramate for migraine prevention in adults, there was no difference in the reduction of migraine frequency based on presence of aura, although it was reported that topiramate reduced the number of migraine auras [89]. In a pediatric study, “visual symptoms” did not predict response rate to preventative treatment with topiramate [90].

Historically, daily aspirin was used in MA with the hope that it would diminish the increased risk of ischemic stroke associated with this disorder—a question that still remains unanswered. In adults with MA, daily aspirin reduced aura frequency [91]. In pediatric patients, there has been one clinical trial examining daily aspirin compared to flunarizine for migraine prevention which showed no difference between the two groups, but both were effective at reducing headache frequency [92]. This study did not examine the MA subgroup separately and did not include a placebo group. In a post-hoc analysis in adult women with MA treated with daily aspirin, there was an increased risk of myocardial infarction [93]. While this may be a chance finding, further studies are needed before recommending daily aspirin for migraine prevention.

Botulinum toxin injection is recommended for use in adults with chronic migraines. Interestingly, there are some data to suggest better efficacy in MA. Adults who have MA were more likely to respond to rimabotulinumtoxinB injections compared to those who have MO [94]. In a separate study examining the use of onabotlinumtoxinA injections in adults with HM, onabotlinumtoxinA injections reduced both the frequency and severity of the migraine headache as well as the frequency and severity of the aura [95]. In adolescents, onabotulinumtoxinA injections are not FDA approved but have been used off-label for migraine prevention. Although there are many case series and retrospective analyses that have demonstrated the benefit of botulinum toxin use in adolescents with chronic migraine, in a randomized trial in this population, there was no difference in effect between the placebo and treatment group [96]. Further studies focusing on MA specifically are needed in this population.

### 7.3. Unique Treatment Considerations for Hemiplegic Migraine and Migraine with Brainstem Aura

HM and migraine with brainstem aura are often thought of as more severe migraine disorders given their disabling features (e.g., weakness, alterations in consciousness) and, in HM, the longer duration of aura symptoms [49]. Historically, there has been concern that use of triptans in these patients could worsen or prolong these disabling auras through intracranial vasoconstriction, so their use has not been recommended. However, more recent studies show that triptans likely do not cause intracranial vasoconstriction and are safe and effective in HM and migraine with brainstem aura [97,98,99]. In a small retrospective study in pediatric patients, triptans were prescribed to patients with HM, migraine with brainstem aura, and retinal migraine, and no adverse effects were observed [36]. In a practice guideline update from the AAN and the AHS, it is noted that the contraindication to use of triptans in HM and migraine with brainstem aura was “based on a view of migraine pathophysiology that is no longer considered current” [50]. These guidelines suggest, however, that larger studies are needed in order to officially modify these recommendations.

Because HM is associated with gene mutations causing channelopathies, preventative medications targeting ion channels have been proposed as particularly useful in this migraine subtype. Flunarizine and verapamil are calcium channel blockers, and there is some evidence supporting their efficacy in treating HM. While flunarizine is not licensed in the United States, it is widely used in Europe for migraine prevention. In a retrospective cohort study, 57% of children had a reduction in headache frequency with flunarizine use, and when HM was evaluated independently, 85% of children with HM had a reduction in attacks [100]. There are adult case reports and small studies supporting the use of verapamil in HM, but this has yet to be studied in the pediatric population [101,102]. Acetazolamide inhibits carbonic anhydrase and may be effective at reducing HM attacks [101]. Within MA subtypes, HM has the most evidence for treatments targeted at auras. As a result, these data are often extrapolated for use in patients who have particularly bothersome typical auras.

### 7.4. Newer Migraine Targeted Therapies

CGRP antagonists have recently been approved for prevention of MA in adults. There are currently four large molecule monoclonal antibodies directed against the CGRP receptor or ligand and approved by the FDA for prevention of migraine with and without aura in adults, including erenumab (subcutaneous injection), fremanezumab (subcutaneous injection), galcanezumab (subcutaneous injection), and eptinezumab (intravenous injection). There is one small-molecule CGRP receptor antagonist—atrogepant (oral)—that is FDA-approved for both MA and MO prevention in adults [103,104,105,106,107,108,109,110,111,112]. For acute migraine attacks with and without aura in adults, there are two oral FDA-approved small-molecule CGRP antagonists (also referred to as “gepants”), rimegepant and ubrogepant [113,114,115]. While trials of these medications included adults with both MA and MO, none of the results were stratified by the presence of aura. Given that blocking CGRP reduces CSD, it could be hypothesized that CGRP antagonists may be more effective in MA, but additional studies are needed. CGRP antagonists have not been approved for use in children or adolescents, though trials are in progress.

## 8. Conclusions

While there is some overlap in the pathophysiology, clinical characteristics, and treatment between MA and MO, there are important distinctions that support MA being a unique entity. Specifically, the pathophysiology of MA is understood to be due to CSD. While the exact pathophysiology of MO is less certain, there is evidence that CSD may not play as critical of a role. CSD is believed to produce the focal neurological signs that are the hallmark of auras and help to distinguish MA from MO. MA also has unique co-morbidities that are not seen in MO. Most notable is the increased stroke risk, which further emphasizes a likely difference in basic pathophysiology between these two entities.

While treatment recommendations are often similar between MA and MO, when studies are stratified by aura, difference in treatment efficacy can be seen. For acute therapies, triptans may be less efficacious in MA, while intravenous magnesium and TMS may be more efficacious in MA. For preventative therapies, lamotrigine and botulinum toxin injections may be more efficacious in MA. Daily aspirin for preventative migraine therapy remains controversial but may reduce MA frequency. Specifically, for HM, flunarizine, verapamil, and acetazolamide may be more efficacious in treating this disorder compared to MO. The newest class of targeted migraine therapies—CGRP antagonists—have not specifically examined efficacy in MA.

Given these differences, larger pediatric studies stratified by the presence of aura are needed to better understand MA and the best management strategies in this age group.

## Figures and Tables

**Table 1 children-08-00228-t001:** ICHD-3 criteria for migraine with aura.

A. At least two attacks fulfilling criteria B and C
B. One or more of the following fully reversible aura symptoms: 1. visual 2. sensory 3. speech and/or language 4. motor 5. brainstem 6. retinal
C. At least three of the following six characteristics: 1. at least one aura symptom spreads gradually over ≥5 min 2. two or more aura symptoms occur in succession 3. each individual aura symptom lasts 5–60 min ^1^ 4. at least one aura symptom is unilateral ^2^ 5. at least one aura symptom is positive ^3^ 6. the aura is accompanied, or followed within 60 min, by headache
D. Not better accounted for by another ICHD-3 diagnosis.

ICHD-3: International Classification of Headache Disorders version 3; ^1^ When, for example, three symptoms occur during an aura, the acceptable maximal duration is 3 × 60 min; motor symptoms may last up to 72 h. ^2^ Aphasia is always regarded as a unilateral symptom, dysarthria may or may not be. ^3^ Scintillations and pins and needles are positive symptoms of aura.

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
