# Peer review of "Pediatric Episodic Migraine with Aura: A Unique Entity?"

_children, 2021, doi:10.3390/children8030228_

Round 1

Reviewer 1 Report

Review

Pediatric Migraine with Aura: A Unique Entity?

This is an excellent and comprehensive review.  In clinical practice, there is discomfort with aura when is accompanies migraine.  This was a comprehensive review that includes how to deceiver mimics, what is known about pathophysiology and treatments.  The authors made a significant effort to include pediatric studies to make this a pediatric focused review.  

Note to authors: Great Review- Glad you included section on mimics. No recommended edits.

Author Response

This is an excellent and comprehensive review.  In clinical practice, there is discomfort with aura when is accompanies migraine.  This was a comprehensive review that includes how to deceiver mimics, what is known about pathophysiology and treatments.  The authors made a significant effort to include pediatric studies to make this a pediatric focused review.  

Note to authors: Great Review- Glad you included section on mimics. No recommended edits.

Thank you for the comments.

Reviewer 2 Report

Review of Manuscript 1127529:  Pediatric Migraine with Aura:  A Unique Entity

This is a well presented and well referenced review.  The review suggests that Pediatric Migraine with Aura may be a unique entity and may require some differences in the therapeutic approach from migraine without aura. Several comments that should be addressed are below.

  1. The data on migraine over the life span should be briefly presented as shown by Lipton and others. These population data show that in prepubertal children the prevalence of migraine is around 5% and following puberty there is a rise in migraine to approximately 30% by age 30, remaining at that level from age 30 to 50-60 and then a decline back to around 3-5% after age 70.  These data suggest that the pediatric age group should be considered as two subgroups with the prepubertal subjects as one and post-pubertal subjects (age puberty to 18-20) as the second.  Is the prevalence of migraine with aura (MA) the same in both of these?  There may be no or minimal data available.  If so this should be stated.  If the data is available, then the data for the two groups should be presented.
  2. Do prepubertal children with migraine without Aura evolve to MA during the postpubertal period, and if so does this change the evaluation and therapeutic approach.
  3. Do pediatric patients with MA evolve into adult migraineurs with MA?
  4. The Early papers on MA by Olesen’s group found that visual aura was associated with MA 99% of the time. Is the same pattern seen in children?  Is occurrence of nonvisual aura symptoms without the visual aura reported?
  5. Post-traumatic headache often manifests with the phenotype of migraine. Is there any relation between head injury (traumatic brain injury) and Pediatric MA?
  6. Journal title is absent from several citations in the references (such as #15,51,69,79). Is this intentional, and  if so , why?

Reviewer 3 Report

This paper deals with a relevant topic.

I have a few suggestions for possible improvements.

1) Avoid citations in the abstract.

2) Exclude Hemiplegic Migraine from the paper, since it's a definite and peculiar condition (significantly different from MA).

3) As to preventive treatment, literature is not examined in sufficient detail (evidence concerning most drugs is ignored, negative findings regarding lamotrigine are not reported).

4) Nothing is said about the generale management of MA/MO and about the psychological aspects, despite these being the most relevant options for preventions according to Cochrane reviews.

5) The use a pain-killer on daily basis should be avoided to prevent MOH (this is not discussed, but should be).

6) Nothing is said about MOH and CDH.

Round 2

Reviewer 2 Report

Re-review: 3/11/2021

Comments to Authors:

This is an excellent review of Pediatric Migraine with aura.  I have only one question to offer

The ICDH stipulates that Aura should last from 5-60 minutes.  The authors have mentioned that aura may continue longer.  There are occasional instances in adults where patients report visual symptoms compatable with aura that last 1-4 minutes or may have repeated brief visual aural-like symptoms for 1-2 minutes repeatedly at the peak of migraine pain and these may re-occur in other subsequent headaches. .  Are these seen in children?  If so how would the authors classify them?  Are these Aura or something else? 

Reviewer 3 Report

I thank the authors for reading and providing meaningful answers to my previous comments.

I can see they worked hard to improve their paper and I think that they reached their goal.

My only final concern is that I see referenced a number of quality studies that have nothing to do with children. I think that looking for similar papers conducted in children and adolescents, if and when available, could further improve the paper.
